# Beyond Scalar Rewards:
# An Axiomatic Framework for Lexicographic MDPs

**Mehran Shakerinava**[1 2 *]**, Siamak Ravanbakhsh**[1 2]**, Adam Oberman**[2 3 4]
[1]School of Computer Science, McGill University [2]Mila – Quebec AI Institute
[3]Department of Mathematics and Statistics, McGill University [4]LawZero

## Abstract

Recent work has formalized the reward hypothesis through the lens of expected utility theory, by interpreting reward as utility. Hausner's foundational work showed that dropping the continuity axiom leads to a generalization of expected utility theory where utilities are *lexicographically* ordered *vectors* of arbitrary dimension. In this paper, we extend this result by identifying a simple and practical condition under which preferences in a Markov Decision Process (MDP) cannot be represented by scalar rewards, necessitating a 2-dimensional reward function. We provide a full characterization of such reward functions, as well as the general $d$-dimensional case under a memorylessness assumption on preferences. Furthermore, we show that optimal policies in this setting retain many desirable properties of their scalar-reward counterparts, while in the Constrained MDP (CMDP) setting – another common multiobjective setting – they do not.

## 1 Introduction

Framing decision-making as an optimization problem typically begins with specifying one or more utility functions and defining an objective with respect to these utilities. The reward hypothesis of Reinforcement Learning (RL) advocates for "maximization of the expected value of the cumulative sum of a received scalar signal" (Sutton, 2004; Sutton & Barto, 2018; Littman, 2017). Other approaches include specifying several utility functions with the objective of finding a solution whose expected utilities are Pareto optimal. In the Constrained Markov Decision Process (CMDP) framework (Altman, 1999), the goal is to maximize the expectation of a primary utility subject to constraints on the expected values of auxiliary utilities. Another approach is *lexicographic optimization*, where utilities are ordered by priority, and the objective is to lexicographically maximize the expected utility vector (Gábor et al., 1998).

It is common to pick one such objective based on heuristics, and then study its properties and propose algorithms for it. *The axiomatic approach goes in the opposite direction* and characterizes a family of utility functions and an objective that correspond to a set of assumptions (axioms) on our preferences. In other words, it shows the *sufficiency* of an objective in capturing certain properties. For instance, it is well-known that a lexicographic order cannot be scalarized, *e.g.,* an order-preserving mapping from a lexicographically ordered square to a line is not possible even though both sets have the same cardinality. In the axiomatic expected utility framework, one can show that a utility function that cannot be scalarized can always be captured by a lexicographic utility function (Hausner, 1954).

In this work, motivated by potential applications in AI safety, we show how this axiomatic approach can lead to lexicographic objectives in MDPs. A lexicographic objective assumes that an infinitesimal increase in the first-priority objective is preferred to any amount of increase in the second-priority objective. Such settings might appear when there is a critical safety requirement that should be

---

*Correspondence to <mehran.shakerinava@mail.mcgill.ca>. Work done in part while interning at LawZero.

39th Conference on Neural Information Processing Systems (NeurIPS 2025).

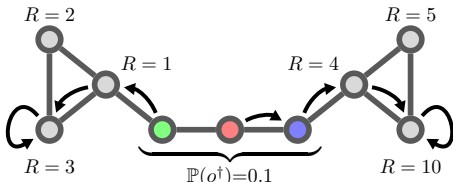

Figure 1: An example of lexicographic planning with time-horizon $T = 4$ steps. The diagram depicts a $0.1$ probability of unsafety when transitioning from a green, red, or blue state. It also depicts the reward from gray states as $R$ and the optimal policy as black arrows. Starting from the green state, the optimal policy is to forgo the high reward on the right side in favor of increased safety and go left. Starting from the red state, since the safety of going left and right is the same, the optimal policy is to go right to get more cumulative reward. Importantly, decisions have to be based on *two* quantities: the probability of safety and expected cumulative reward.

prioritized above all else. Another common and practical example is when the first priority is to achieve some goal while the second priority is to minimize the time that it takes. Lexicographic objectives are also reminiscent of Asimov's Three Laws of Robotics (Asimov, 1942) where the highest priority objective is to "not injure a human being or, through inaction, allow a human being to come to harm." An illustrative example of a lexicographic objective is presented in Fig. 1.

## 1.1 Outline and Summary of Results

We begin by reviewing the relevant literature (Section 2), followed by a brief summary of von Neumann-Morgenstern (vNM) expected utility theory and Hausner's lexicographic extension of it (Section 3). We then extend Hausner's result to the sequential decision-making setting (Section 4), which involves formally specifying a setting where outcomes are sequences of abstract events, and introducing an axiom to structure preferences. Since Hausner's theorem does not specify the dimensionality of utility vectors, we study a simple setting that naturally leads to a 2-dimensional lexicographic utility function (Section 5), and extend this to the sequential setting (Section 6). This yields a simple and interpretable recursive equation for the utility of event sequences. We then study the properties of optimal policies in lexicographic MDPs, highlighting similarities with scalar-reward MDPs and contrasting them with the CMDP framework (Section 7). We then further compare LMDPs and CMDPs (Section 8). We conclude with a discussion of limitations and future work (Section 9) and concluding remarks (Section 10).

## 2 Related Work

**Expected Utility Theory.** Expected utility theory originated with Bernoulli (1738) and was formalized axiomatically by von Neumann & Morgenstern (1947) in the context of game theory. This axiomatic approach characterizes when and why maximizing expected utility is justified. Subsequent work refined these foundations and extended them to more general settings, such as alternative formulations of the axioms (Jensen, 1967) and generalization from finite sets to mixture spaces (Herstein & Milnor, 1953). A comprehensive account is provided by Fishburn (1982), which also contains a relevant chapter on lexicographic expected utility, due to Hausner (1954). In the lexicographic setting, reducing the dimensionality of utility vectors requires additional assumptions, which are often technical and unintuitive. Our work contributes a natural and interpretable condition that leads to 2-dimensional lexicographic utilities.

**Sequential Decision-Making and Expected Utility Theory.** Classical expected utility theory typically applies only to final outcomes, abstracting away the sequential nature of decision-making and preventing utility assignment to individual interactions, as is common in MDP and RL settings. Recent works have extended vNM rationality axioms to sequential decision-making, motivated by variable discount factors (Pitis, 2019) and formalizing the reward hypothesis (Shakerinava & Ravanbakhsh, 2022; Bowling et al., 2023). Following this axiomatic approach, we extend lexicographic expected utility theory to sequential settings, deriving structured reward functions under minimal and interpretable assumptions.

**Lexicographic Decision-Making.** Lexicographic optimization in Multi-Objective Reinforcement Learning (MORL) has been previously studied, notably by Gábor et al. (1998), and extended in Skalse et al. (2022), which present a family of both action-value and policy gradient algorithms for lexicographic RL with theoretical convergence results and empirical performance on benchmarks. In other works, Wray et al. (2015) provide a lexicographic value iteration algorithm and prove convergence under the assumption that each objective is allowed some acceptable level of slack. El Khalfi (2017) propose algorithms for finding a lexicographic optimal policy in possibilistic MDPs. Hahn et al. (2021) investigate lexicographic $\omega$-regular objectives within formal verification, proposing a reduction that enables model-free RL for prioritized temporal logic specifications.

**Multi-Objective Decision-Making and AI Safety.** The standard RL paradigm assumes that a single scalar reward is sufficient for specifying goals (Silver et al., 2021). This view has been challenged by Vamplew et al. (2022), who argue that in safety-critical settings, scalar rewards can incentivize unsafe or undesirable behaviors. Lexicographic objectives have been proposed in domains such as autonomous vehicles, where safety must take strict precedence over other considerations (Zhang et al., 2022). Omohundro (2008) warns that unbounded reward maximization may give rise to dangerous instrumental drives. While scalarization methods are commonly used to combine multiple objectives, they suffer from well-known theoretical limitations (Vamplew et al., 2008). Multi-objective RL methods (Hayes et al., 2022) avoid scalarization but remain algorithmically and theoretically more challenging than the scalar case. Complementary to our axiomatic treatment, Miura (2023) analyzes the expressivity of scalar vs. multi-dimensional rewards and gives necessary and sufficient conditions for when a specified set of acceptable policies can be realized.

## 3 Background

### 3.1 Scalar Expected Utility Theory

The expected utility theory of von Neumann & Morgenstern (1947) provides an axiomatic treatment of decision-making under uncertainty. Let $\mathcal{O}$ denote the set of outcomes. The result of a decision is always an element from this set. We refer to a distribution over outcomes as a *lottery* and we denote the space of lotteries as $\Delta(\mathcal{O})$. The player is faced with a number of such lotteries and has to pick one. After a lottery has been selected, the player receives an outcome sampled according to the lottery's distribution.

The player supplies their preferences in the form of a relation on the space of lotteries $(\succsim, \Delta(\mathcal{O}))$. Based on this relation, we can define another relation $\succ$ as $p \succ q := \text{not}(q \succsim p)$. We also define the relation $\approx$ as $p \approx q := (p \succsim q \text{ and } q \succsim p)$ and we say that $p$ and $q$ are *indifferent*. Its negation is denoted $\napprox$.

We assume that the player's preferences satisfy von Neumann and Morgenstern's four axioms of rationality. These axioms are defined below, and we will refer to them as the vNM axioms.

*Axiom* 1 (Completeness). For all $p, q \in \Delta(\mathcal{O})$,
$$p \succsim q \text{ or } q \succsim p.$$

*Axiom* 2 (Transitivity). For all $p, q, r \in \Delta(\mathcal{O})$,
$$p \succsim q \text{ and } q \succsim r \implies p \succsim r.$$

*Axiom* 3 (Independence). For all $p, q, r \in \Delta(\mathcal{O})$ and $\alpha \in [0, 1)$,
$$\alpha p + (1 - \alpha)q \succsim \alpha p + (1 - \alpha)r \iff q \succsim r.$$

The lottery $\alpha p + (1 - \alpha)q$ that appears in the statement of Independence can be thought of as a compound lottery constructed by first tossing a biased coin with probability $\alpha$ of landing heads. The outcome of the coin toss determines whether we get an outcome sampled from lottery $p$ (heads) or $q$ (tails). Independence states that comparing compound lotteries constructed with the same biased

coin and with the same lottery for heads, is equivalent to comparing the corresponding lotteries for tails. From an algebraic standpoint, Independence can be thought of as a cancellation law for the comparison of lotteries.

*Axiom* 4 (Continuity). For all $p, q, r \in \Delta(\mathcal{O})$,

$$p \succsim q \succsim r \implies \exists \alpha \in [0,1], \; \alpha p + (1 - \alpha)r \approx q.$$

The Continuity axiom essentially states that, as the probabilities of a lottery vary, our valuation of the lottery changes smoothly. As we will see, it is responsible for the sufficiency of *scalar* utilities.

Before presenting the expected utility theorem, we will need to define utility and what it means for a utility function to be linear.

**Definition 1.** A utility function for $(\succsim, \Delta(\mathcal{O}))$ is any function $u : \Delta(\mathcal{O}) \to \mathbb{R}$ such that, for all $p, q \in \Delta(\mathcal{O})$,

$$u(p) \geq u(q) \iff p \succsim q. \tag{1}$$

**Definition 2.** A utility function is said to be linear if for all $p \in \Delta(\mathcal{O})$,

$$u(p) = \sum_{o \in \mathcal{O}} p(o)u(o).^2 \tag{2}$$

We are now ready to state the von Neumann-Morgenstern (vNM) expected utility theorem.

**Theorem 1** (von Neumann & Morgenstern (1947)) *A relation $(\succsim, \Delta(\mathcal{O}))$ satisfies the vNM axioms if and only if there exists a linear utility function $u : \Delta(\mathcal{O}) \to \mathbb{R}$. Moreover, $u$ is unique up to positive affine transformations, i.e., $u \mapsto au + b$, where $a > 0$.*

For proofs, see von Neumann & Morgenstern (1953), Fishburn (1982), or Maschler et al. (2013).

The vNM theorem identifies settings where decision-making can be optimized by maximizing the expected value of a scalar utility function. In other words, the vNM theorem is the formal basis of the well-known maximum expected utility principle.

We mention two advantages of expected utility theory over other[3] methods for specifying rewards. First, it separates reward specification from the mechanics of the environment, which is particularly useful when the environment is complex or unknown. Second, it enables reliable comparison of suboptimal policies. These advantages, among others, make it an ideal candidate for reward specification.

## 3.2 Lexicographic Expected Utility Theory

An important result due to Hausner (1954) is that, if we forgo Continuity, lotteries can still be compared in terms of expected utility, but the utilities will be vectors and the comparison will be lexicographic. In a lexicographic comparison, entries are compared from first to last and the first differing entry determines the order. The rest of the entries are irrelevant. The mathematical definition is as follows.

**Definition 3.** For all $u, v \in \mathbb{R}^d$,

$$u >_{\text{lex}} v := (\exists k \in [d], u_1 = v_1, ..., u_{k-1} = v_{k-1}, \text{ and } u_k > v_k). \tag{3}$$

To present the lexicographic expected utility theorem we first define what a lexicographic utility function is.

---

[2]We will occasionally abuse notation by writing $o$ (an element of $\mathcal{O}$) when we actually mean the Dirac delta distribution centered at $o$ (an element of $\Delta(\mathcal{O})$). This simplification will typically occur when adding an outcome to a lottery or evaluating its utility.

[3]For example, assigning a reward of 1 to optimal actions and 0 to all other actions or ad-hoc reward shaping methods. More generally, there are many reward functions that can lead to the same optimal policy. A simple case: there are 3 actions $a, b, c$. We prefer $a \succ b \succ c$ but we use a reward function that satisfies $r(a) > r(c) > r(b)$ – notice that $b$ and $c$ are swapped. The optimal action is the same in both cases. Suppose the initial policy selects action $b$ and then we optimize it (w.r.t. rewards) to $c$. However, due to "bad" reward function, the policy actually gets worse. On the other hand, with vNM rewards, increasing expected reward is guaranteed to result in a more preferred policy.

**Definition 4.** A lexicographic utility function for $(\succsim, \Delta(\mathcal{O}))$ is any function $u : \Delta(\mathcal{O}) \to \mathbb{R}^d$ such that, for all $p, q \in \Delta(\mathcal{O})$,

$$u(p) \geq_{\text{lex}} u(q) \iff p \succsim q. \tag{4}$$

A *linear* lexicographic utility function is defined in the same way as Definition 2 with addition and scalar multiplication interpreted as operations on a vector space.

We let $\mathcal{L}_+^{d \times d}$ be the set of $d \times d$ lower triangular matrices with positive diagonal entries, as formally described by Eq. (5). This set will be used in the specification of affine transformations that retain lexicographic ordering.

$$\mathcal{L}_+^{d \times d} := \left\{ A \in \mathbb{R}^{d \times d} \;\middle|\; A_{ij} = 0 \text{ for } i < j, \; A_{ii} > 0 \text{ for all } i \in [d] \right\} \tag{5}$$

We now state Hausner's lexicographic expected utility theorem.

> **Theorem 2** (Hausner (1954)) *A relation $(\succsim, \Delta(\mathcal{O}))$ satisfies Completeness, Transitivity, and Independence if and only if there exist $d \in \mathbb{N}$ and a $d$-dimensional linear lexicographic utility function $u : \Delta(\mathcal{O}) \to \mathbb{R}^d$. Moreover, $u$ is unique up to transformations of the form $u \mapsto Au + b$, where $A \in \mathcal{L}_+^{d \times d}$ and $b \in \mathbb{R}^d$.*

For proofs see Hausner (1954) or Fishburn (1982).

Note that $d = 1$ captures the setting where Continuity is satisfied and the vNM theorem applies, whereas $d > 1$ captures settings where Continuity is violated. Remarkably, the theorem not only recovers the well-known fact that lexicographic orders cannot be scalarized, but also shows that, within the expected utility framework, any non-scalarizable utility function must admit a lexicographic representation.

## 4 Sequential Lexicographic Expected Utility

We now extend lexicographic expected utility to the sequential decision-making setting. In this setting, an agent sequentially collects utility instead of only receiving an outcome with some utility at the end of the interaction. It is common to use the term reward instead of utility, so we will be using these terms interchangeably.

We model the agent's interaction with the environment as a Markov Decision Process (MDP) with the modification that the agent observes events from a set $\mathcal{E}$ instead of rewards. More formally, we assume that the MDP is equipped with a conditional probability distribution $\mathbb{P} : \mathcal{S} \times \mathcal{A} \to \Delta(\mathcal{S} \times \mathcal{E})$ of next state and event given current state and action. This setting is very general since events are allowed to be any stochastic function of current state, action, and next state. The set of outcomes in this setting corresponds to sequences of events, *i.e.*, $\mathcal{O} = \mathcal{E}^* := \{\varepsilon\} \cup \bigcup_{i \in \mathbb{N}} \mathcal{E}^i$, where $\varepsilon$ is the empty sequence. Rewards emerge as a result of applying expected utility theory to a given preference relation on distributions over sequences of events $(\succsim, \Delta(\mathcal{E}^*))$.

Without any additional assumptions, the preference relation is unstructured, overlapping sequences are treated as entirely independent entities, and the utility of any two sequences is independently determined. We will, therefore, introduce a simple and reasonable axiom and show that it leads to utility functions that take a simple mathematical form.

But first, we need to introduce a *concatenation operator* $\cdot$ that will be used in the axiom. We will use it both for concatenating sequences and for concatenating a sequence to a distribution over sequences. The operator is best described with examples.

*Example* 1. Let $(a_1, a_2, a_3)$, $(b_1, b_2)$, and $(c)$ be sequences of events. Then, an example of concatenating to a sequence is $(c) \cdot (b_1, b_2) = (c, b_1, b_2)$. And an example of concatenating to a distribution over sequences is

$$c \cdot \left( \tfrac{1}{3}(a_1, a_2, a_3) + \tfrac{2}{3}(b_1, b_2) \right) = \tfrac{1}{3}(c, a_1, a_2, a_3) + \tfrac{2}{3}(c, b_1, b_2). \tag{6}$$

The axiom that we are about to introduce says that preferences are not affected by past events. As a result, the agent can forget past events and only focus on optimizing future events. The axiom is a version of the memorylessness axiom from Shakerinava & Ravanbakhsh (2022).

*Axiom* 5 (Memorylessness). For all $e \in \mathcal{E}$, either

$$\forall p, q \in \Delta(\mathcal{E}^*), \quad e \cdot p \succsim e \cdot q \iff p \succsim q, \tag{7}$$

or

$$\forall p, q \in \Delta(\mathcal{E}^*), \quad e \cdot p \approx e \cdot q. \tag{8}$$

Events that satisfy Eq. (8) act as *terminal* events since future events no longer have any effect on the final outcome. We denote by $\mathcal{E}_{\text{term}}$ the set of such events.

**Theorem 3** (Sequential Lexicographic Expected Utility Theorem) *A relation $(\succsim, \Delta(\mathcal{E}^*))$ satisfies Completeness, Transitivity, Independence, and Memorylessness if and only if there exist $d \in \mathbb{N}$, a $d$-dimensional linear lexicographic utility function $u$ with $u(\varepsilon) = \mathbf{0}$, rewards $r : \mathcal{E} \to \mathbb{R}^d$, and reward multipliers $\Gamma : \mathcal{E} \to \mathcal{L}_+^{d \times d} \cup \{\mathbf{0}\}$ such that*

$$u(e \cdot \tau) = r(e) + \Gamma(e)u(\tau), \tag{9}$$

*for all $e \in \mathcal{E}$ and $\tau \in \mathcal{E}^*$.*

A proof is provided in Appendix D.1. A point omitted from the theorem's statement, but evident in the proof, is that an event $e$ is terminal if and only if $\Gamma(e) = \mathbf{0}$.

Theorem 3 implies an MDP where rewards $r$ are $d$-dimensional *vectors*, the "discount" factors $\Gamma$ are transition-dependent *matrices* in $\mathcal{L}_+^{d \times d}$, and expected returns are compared *lexicographically*. We refer to such MDPs as Lexicographic MDPs (LMDPs). Notably, the structure of $\Gamma$ is more general than that of previous works (*e.g.,* Skalse et al. (2022)) where it is typically assumed to be diagonal.

The commonly used *transition-independent* discounted rewards emerge as a result of a non-parameterized version of the temporal $\gamma$-indifference from Bowling et al. (2023).

*Axiom* 6 (Temporal $\gamma$-Indifference). For all $e \in \mathcal{E}, \tau_1 \in \mathcal{E}^*, \tau_2 \in \mathcal{E}^*$,

$$\frac{1}{\gamma + 1}(e \cdot \tau_1) + \frac{\gamma}{\gamma + 1}(\tau_2) \approx \frac{1}{\gamma + 1}(e \cdot \tau_2) + \frac{\gamma}{\gamma + 1}(\tau_1),$$

where $\gamma \in [0, 1]$.

**Theorem 4** (Discounted Lexicographic Expected Utility Theorem) *A relation $(\succsim, \Delta(\mathcal{E}^*))$ satisfies Completeness, Transitivity, Independence, and Temporal $\gamma$-Indifference if and only if there exist $d \in \mathbb{N}$, a $d$-dimensional linear lexicographic utility function $u$ with $u(\varepsilon) = \mathbf{0}$, and rewards $r : \mathcal{E} \to \mathbb{R}^d$ such that*

$$u(e \cdot \tau) = r(e) + \gamma u(\tau), \tag{10}$$

*for all $e \in \mathcal{E}$ and $\tau \in \mathcal{E}^*$.*

A proof is provided in Appendix D.2. The proofs of Theorems 3 and 4 are similar to their scalar counterparts in Shakerinava & Ravanbakhsh (2022); Bowling et al. (2023) where we have lifted the continuity axiom and arrived at a reward structure where rewards are vectors and (scalar) comparisons have been replaced with lexicographic comparisons.

## 5 A Single Unsafe Utility

The lexicographic expected utility theorem has a drawback in that it does not specify the dimensionality of the utility function. It also does not specify any particular subspace of $\mathbb{R}^d$ for the utility of outcomes. We would like to have more fine-grained understanding of the structure of the utility function for specific settings of interest.

Therefore, in this section, we propose an intuitive axiom that results in a simple 2-dimensional linear lexicographic utility function. We do so by introducing an outcome $o^\dagger$ that is 'infinitely bad,' as formalized by the next axiom. We interpret this outcome as a critically unsafe outcome. Let $\mathcal{O}^\dagger := \left\{ o \mid o \approx o^\dagger \right\}$ and $\mathcal{O}_{\text{safe}} := \mathcal{O} - \mathcal{O}^\dagger$.

*Axiom* 7 (Safety First). For all $p, q \in \Delta(\mathcal{O}_{\text{safe}})$ and all $\varepsilon > 0$,
$$\varepsilon o^\dagger + (1 - \varepsilon)p \prec q.$$

The existence of such an outcome violates Continuity. To see that, let $p, q \in \Delta(\mathcal{O}_{\text{safe}})$ be two safe lotteries such that $p \succ q$. We now have $p \succ q \succ o^\dagger$. Then, $\alpha p + (1 - \alpha)o^\dagger \prec q$ for all $\alpha < 1$ and $\alpha p + (1 - \alpha)o^\dagger \succ q$ for $\alpha = 1$. For no $\alpha$ do the two sides become indifferent, so Continuity is violated. Therefore, from now on, we assume that Continuity only holds for $(\succ, \Delta(\mathcal{O}_{\text{safe}}))$. The full set of assumptions are provided below.

**Assumption 1.** The relation $(\succsim, \Delta(\mathcal{O}))$ satisfies Completeness, Transitivity, and Independence and the relation $(\succsim, \Delta(\mathcal{O}_{\text{safe}}))$ satisfies Continuity. The relation $(\succsim, \Delta(\mathcal{O}))$ and $o^\dagger$ satisfy Safety First. Also, there exist $o_1, o_2 \in \mathcal{O}_{\text{safe}}$ such that $o_1 \not\approx o_2$ (non-triviality).

In this setting, lotteries can be uniquely written in the form
$$[\alpha, p] := (1 - \alpha)o^\dagger + \alpha p, \tag{11}$$
where $\alpha \in [0, 1]$ is the probability of safety and $p \in \Delta(\mathcal{O}_{\text{safe}})$. An exception occurs when $\alpha = 0$ which can be addressed by fixing a lottery $q \in \Delta(\mathcal{O}_{\text{safe}})$ and writing the (deterministic) lottery $o^\dagger$ as $[0, q]$, making its representation unique.

*Example* 2. The lottery $p = {}^1\!/_3\, o^\dagger + {}^1\!/_2\, x + {}^1\!/_6\, y$ means there is a ${}^1\!/_3$ chance of obtaining outcome $o^\dagger$, a ${}^1\!/_2$ chance of obtaining outcome $x$, and a ${}^1\!/_6$ chance of obtaining outcome $y$. It is uniquely decomposed into the form of Eq. (11) as
$$p = \left(1 - \frac{2}{3}\right)o^\dagger + \frac{2}{3}\left(\frac{3}{4}x + \frac{1}{4}y\right) = [\frac{2}{3}, \frac{3}{4}x + \frac{1}{4}y].$$

The next lemma shows that it is possible to compare any two lotteries first by comparing the probability of safety and, if equal, performing a comparison in $\Delta(\mathcal{O}_{\text{safe}})$.

**Lemma 1.** *For all $p, q \in \Delta(\mathcal{O})$ and all $\alpha, \beta \in [0, 1]$,*
$$[\alpha, p] \succsim [\beta, q] \iff (\alpha > \beta) \text{ or } (\alpha = \beta \text{ and } p \succsim q). \tag{12}$$

A proof is provided in Appendix D.3.

With the appropriate definitions and Lemma 1 at hand, we are now ready to prove the following theorem.

**Theorem 5** (Lexicographic Expected Utility Theorem with a Single Unsafe Utility) *A relation $(\succsim, \Delta(\mathcal{O}))$ satisfies Assumption 1 if and only if there exist a linear utility function $u' : \Delta(\mathcal{O}_{\text{safe}}) \to \mathbb{R}$ for $(\succsim, \Delta(\mathcal{O}_{\text{safe}}))$ and a 2-dimensional linear lexicographic utility function $u : \Delta(\mathcal{O}) \to \mathbb{R}^2$ such that for all $o \in \mathcal{O}$,*
$$u(o) = \begin{cases} (0, u'(o)) & o \in \mathcal{O}_{\text{safe}}, \\ (-1, 0) & o \approx o^\dagger. \end{cases} \tag{13}$$

*Moreover, $u$ is unique up to transformations of the form $u \mapsto Au + b$, where $A \in \mathcal{L}_+^{2 \times 2}$ and $b \in \mathbb{R}^2$.*

A proof is provided in Appendix D.4.

*Example* 3. A linear lexicographic utility function with a single unsafe utility is depicted in Fig. 2.

In the utility function of Eq. (13), the first dimension of the utility of a lottery represents the probability of safety minus 1 and the second dimension represents the expected utility given safety. Also note that it is 2-dimensional, as opposed to Theorem 2 which does not specify $d$.

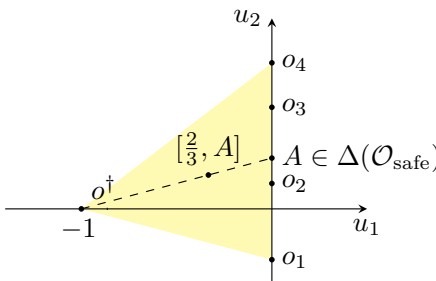

Figure 2: An example plotting the 2-dimensional lexicographic utility of $\mathcal{O} = \{o_1, ..., o_4, o^\dagger\}$. Assuming $u$ is linear, its range will be the highlighted triangle. Note that Continuity is the main axiom responsible for utilities being mapped onto a straight line. Since $o^\dagger$ does not satisfy Continuity, it gets mapped outside the line containing the set of points $u(\mathcal{O}_{\text{safe}})$.

## 6 Sequential Lexicographic Expected Utility with a Single Unsafe Utility

We now extend the single unsafe utility setting to the sequential setting according to the framework introduced in Section 4. We define $\mathcal{E}^\dagger$ as the events that are indifferent to $o^\dagger$. Adding the Memorylessness axiom to our previous assumptions leads to the following theorem.

**Theorem 6** (Sequential Lexicographic Expected Utility Theorem with a Single Unsafe Utility) *A relation $(\succsim, \Delta(\mathcal{E}^*))$ satisfies Assumption 1 and Memorylessness if and only if there exist rewards $r : \mathcal{E} \to \mathbb{R}$, reward multipliers $\gamma : \mathcal{E} \to \mathbb{R}_+$, and 2-dimensional linear lexicographic utility function $u : \Delta(\mathcal{E}^*) \to \mathbb{R}^2$ satisfying $u(\varepsilon) = \mathbf{0}$ and*

$$u(e \cdot \tau) = \begin{cases} (0, r(e)) & e \in \mathcal{E}_{\text{term}} - \mathcal{E}^\dagger \\ (-1, 0) & e \in \mathcal{E}^\dagger \\ (u_1(\tau), (1 + u_1(\tau))r(e) + \gamma(e)u_2(\tau)) & \text{otherwise} \end{cases} \tag{14}$$

*for all events $e \in \mathcal{E}$ and sequences of events $\tau \in \mathcal{E}^*$.*

A proof is provided in Appendix D.5.

It follows from Theorem 6 that the events $\mathcal{E}^\dagger$ are terminal and result in an unsafe outcome $\approx o^\dagger$. It is worth noting that Eq. (14) can be somewhat simplified by assuming that terminal events lead to virtual terminal states. Observe that letting $u(\tau) = (0,0)$ in the third case of Eq. (14) produces the first case and letting $u(\tau) = (-1, 0)$ produces the second case. Therefore, we can mainly use the third case and obtain the other two cases by assigning a utility of $(0,0)$ to safe terminal states and a utility of $(-1, 0)$ to unsafe terminal states. We also learn from Theorem 6 that each event $e$ has a corresponding reward multiplier $\gamma(e) > 0$. By assuming Temporal $\gamma$-Indifference on $(\succsim, \Delta(\mathcal{O}_{\text{safe}}))$ one can arrive at a fixed transition-independent discount factor $\gamma \in (0, 1]$ for the second dimension.

*Example* 4. Fig. 1 depicts an example of the described setting. For simplicity, we are assuming that the agent can deterministically move to a neighboring state and that events are a stochastic function of the starting state of any transition.

Eq. (14) can also be written in the general form of Eq. (9) with $d = 2$. Let us refer to $r$ and $\Gamma$ from Eq. (9) as $\tilde{r}$ and $\tilde{\Gamma}$. Specifying them as follows recovers Eq. (14).

$$\tilde{r}(e) := \begin{cases} (0, r(e)) & e \in \mathcal{E} - \mathcal{E}^\dagger \\ (-1, 0) & \text{otherwise} \end{cases} \tag{15}$$

$$\tilde{\Gamma}(e) := \begin{cases} \begin{pmatrix} 1 & 0 \\ r(e) & \gamma(e) \end{pmatrix} & e \in \mathcal{E} - \mathcal{E}_{\text{term}} \\ \mathbf{0} & \text{otherwise} \end{cases} \tag{16}$$

# 7 Properties of Optimal Policies in Lexicographic MDPs

In this section, we examine the properties of optimal policies in the lexicographic setting, referred to as LMDPs. We highlight their similarities to the scalar MDP setting and contrast them with the CMDP framework.

The first important point of contrast is that in CMDPs, an optimal stationary policy may depend on the starting state distribution (Altman, 1999), whereas in MDPs, there exists a stationary policy that is optimal for all starting states (Puterman, 1994). We refer to this stronger notion of optimality as *uniform optimality*.

Next, we show that the fundamental theorem of MDPs (see Appendix A) still holds in the absence of the continuity axiom. To do so we will need to make the assumption that the diagonal entries of $\Gamma(e)$ are less than 1 for all $e \in \mathcal{E}$. This is analogous to the assumption that the discount factor is less than 1 for MDPs.

**Assumption 2.** For all $e \in \mathcal{E}$ and $i \in [d]$, $\Gamma_{i,i}(e) < 1$.

> **Theorem 7** (Fundamental Theorem of LMDPs) *For every finite LMDP satisfying Assumption 2, a policy $\pi : \mathcal{S} \to \mathcal{A}$ is uniformly optimal if and only if it is greedy w.r.t. $Q^\star$, that is, $\mathbb{E}_{a \sim \pi(s)}[Q^\star(s,a)] = \text{lex} \max_a Q^\star(s,a)$ for all $s \in \mathcal{S}$.*

A proof is provided in Appendix D.6. We write lex max instead of max only to emphasize that the argument is a set of vectors that are compared lexicographically. Also, $Q^\star$ for LMDPs is defined similar to MDPs where the maximization is interpreted as a lexicographic maximization.

**Corollary 1.** *For every finite LMDP satisfying Assumption 2, there exists a stationary deterministic uniformly optimal policy.*

*Proof.* Any policy $\pi$ such that $\pi(s) \in \arg \text{lex} \max_a Q^\star(s,a)$ is a stationary deterministic uniformly optimal policy. $\square$

The result above mirrors the MDP setting but stands in stark contrast to CMDPs. In a CMDP, an optimal policy might need to randomize its actions, and, as mentioned before, a uniformly optimal stationary policy might not exist (Altman, 1999; Szepesvári, 2020). The fundamental reason for these differences is that CMDPs violate both Independence and Continuity (Bowling et al., 2023, §7), whereas LMDPs violate only Continuity. In fact, it has been shown that a slightly weaker notion of Independence is sufficient for guaranteeing the existence of a uniformly optimal stationary policy in trees (Colaço Carr et al., 2024). We conclude that the fundamental properties of optimal policies in MDPs do not rely on the Continuity axiom.

# 8 Further Comparison of LMDP and CMDP

We saw some differences between LMDPs and CMDPs in the previous section. We now provide a more detailed comparison of the two frameworks.

**Can any CMDP be turned into an LMDP?** No. Consider the following example with three outcomes $o_1, o_2, o_3$, two utility functions $u_1, u_2$, and the constrained objective max $\mathbb{E}[u_1]$ subject to $\mathbb{E}[u_2] \geq 0$. The utilities are specified as $u(o_1) = (4, -2), u(o_2) = (2, 2), u(o_3) = (0, 4)$. Now consider the lotteries $A = \frac{1}{2} o_1 + \frac{1}{2} o_3$ and $B = \frac{1}{2} o_2 + \frac{1}{2} o_3$. Assuming linear utilities we have $u(A) = (2, 1)$ and $u(B) = (1, 3)$. Both of these lotteries satisfy the constraint. Since $A$ has higher expected $u_1$ utility it is preferred over $B$. Now apply independence and remove $o_3$ from both lotteries and call the resulting (pure) lotteries $A' = o_1$ and $B' = o_2$. According to independence, $A'$ should be preferred to $B'$. But that is not the case because the outcome $o_1$ does not satisfy the constraint while $o_2$ does, so $B'$ is preferred to $A'$, violating independence. Since LMDPs satisfy independence, CMDPs cannot in general be turned into LMDPs. We note that a similar example is provided by Bowling et al. (2023).

**Can any LMDP be turned into a CMDP?** Only under certain conditions. If we fix the starting state distribution and, given $K$ priority levels, we know the optimal value of all but the lowest priority level, *i.e.,* we know $V_1^\star, ..., V_{K-1}^\star$, then it is possible to turn the LMDP problem into a

CMDP by constraining these $K-1$ values with the known optimal values and optimizing the lowest-priority value, *i.e.*, $\max \mathbb{E}[V_K(s_0)]$ subject to $\mathbb{E}[V_i(s_0)] \geq V_i^\star$ for all $i \in 1, ..., K-1$. Essentially, the first $K-1$ optimization problems are turned into constraints, similar to turning $\max_x f(x)$ into $\max_x 0$ subject to $f(x) \geq f^\star$. For example, if we were to turn an LDMP with a single unsafe outcome into a CMDP, then the CMDP would constrain the probability of the unsafe outcome, assuming access to optimal achievable probability of safety (*i.e.*, $V_1^\star$), rather than optimizing it.

Another important distinction between CMDPs and LMDPs is that CMDPs put constraints on expected outcome while LMDPs only allow constraints on specific outcomes. This means that if one observes a single run of a policy, in the case of an LMDP, one can evaluate this run individually and assign a utility vector to it, but in the case of a CMDP, it is not possible to tell if constraints are satisfied or not, and so it is not possible to evaluate a single run. The root cause of this difference can be traced back to CMDP's violation of the independence axiom.

> *Example* 5. Consider the toy-example of a robot that needs to exit a maze. To incentivize the robot to exit as fast as possible, commonly, we either assign a reward of $-1$ to each step or we use a discount factor and assign a positive reward for escaping. Now consider a maze with hazards that can possibly destroy the robot. Suppose we would like, 1st, to maximize the probability of safely exiting the maze, and 2nd, to do so as fast as possible. Now suppose we have to design a reward function for this task without having seen the maze, *e.g.*, we do not know how likely the hazards are to destroy the robot or how large the maze is. We might try to assign a large negative reward to the robot being destroyed. However, that does not adequately capture our prioritized objective. For a given scalar reward function, we can always design a maze such that the robot takes a more hazardous path by making safe paths longer or hazard probabilities smaller. An LMDP can capture this objective with a 2-dimensional reward function where the reward for a hazardous event is $(-1, 0)$. If we were to use CMDPs for this problem, we would need to know what the optimal hazard probability is in order to set a constraint on it. Additionally, changing the starting state to a point in the maze that has a different optimal hazard probability necessitates respecifying the CMDP.

## 9 Limitations and Future Work

Lexicographic objectives are only appropriate in settings where objectives cannot be traded off. Such cases may be less common in practice. Also, we have not proposed new algorithmic methods for solving lexicographic MDPs or RL problems. Although prior works have developed algorithms in this space (Wray et al., 2015; Skalse et al., 2022), lexicographic optimization remains challenging. Furthermore, our analysis focuses on environment-independent reward specification, following the expected utility theory paradigm. In practice, when the environment is known and fixed, it may be possible to design simpler, scalar rewards that suffice for a specific task. Lexicographic objectives have potential applications in AI safety, particularly in problems of AI control. For example, a primary objective might be to maintain a safety guardrail (Bengio et al., 2025), to ensure that an AI system remains confined to a sandbox environment, or that its influence is restricted in a controlled manner. Exploring such applications is a promising avenue for future research.

## 10 Conclusion

We presented a lexicographic generalization of expected utility theory for sequential decision-making, motivated by settings where objectives must be prioritized in a strict, non-compensatory order. Building on Hausner's extension of expected utility theory, we identified a simple and practical condition under which preferences cannot be captured by scalar rewards, necessitating lexicographically ordered utility vectors. We provided a full characterization of such utility functions in Markov Decision Processes (MDPs) under a memorylessness assumption on preferences, including both the 2-dimensional case and the general $d$-dimensional case. Importantly, we showed that optimal policies in this setting retain key properties of scalar-reward MDPs, such as the existence of stationary, uniformly optimal policies, in contrast to the Constrained MDP (CMDP) framework. Our results generalize the scalar reward hypothesis while preserving the utility-maximization paradigm, offering a principled foundation for lexicographic objectives in sequential decision-making.

## Acknowledgments

We thank Mehrab Hamidi, Motahareh Sohrabi, Sékou-Oumar Kaba, Vedant Shah, Damiano Fornasiere, Gauthier Gidel, Yoshua Bengio, and the LawZero team for feedback on earlier drafts of this paper. This work was in part supported by Canada CIFAR AI Chairs and NSERC Discovery programs.

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

# A Background: Reinforcement Learning

Reinforcement Learning (RL) studies the setting where an agent interacts with an environment while receiving a reward signal in response (Sutton & Barto, 2018). The environment is typically modeled as a Markov Decision Process (MDP). An MDP can be described by a set of states $\mathcal{S}$, set of actions $\mathcal{A}$, transition and reward probabilities $\mathbb{P}(s_{t+1}, r_{t+1} \mid s_t, a_t)$, and a discount factor $\gamma \in [0, 1)$. The objective is to produce a policy $\pi : \mathcal{S} \to \mathcal{A}$ that maximizes expected cumulative discounted reward, that is, $\mathbb{E}_\pi[\sum_{t=0}^{\infty} \gamma^t \mathbf{r}_t]$. The cumulative discounted reward will sometimes be referred to as *return*.

The expected return of starting from state $s$ and following policy $\pi$ is known as the *value function* and is defined as

$$V^\pi(s) := \mathbb{E}_\pi\left[\sum_{t=0}^{\infty} \gamma^t \mathbf{r}_t \;\middle|\; s_0 = s\right]. \tag{17}$$

The expected return of starting from state $s$, taking action $a$, and thereafter following policy $\pi$ is known as the *Q-value function* and is defined as

$$Q^\pi(s, a) := \mathbb{E}_\pi\left[\sum_{t=0}^{\infty} \gamma^t \mathbf{r}_t \;\middle|\; s_0 = s, a_0 = a\right]. \tag{18}$$

The *optimal Q-value function* is defined as

$$Q^\star(s, a) := \max_\pi Q^\pi(s, a). \tag{19}$$

A policy $\pi$ is *optimal* if it satisfies $\mathbb{E}[Q^\pi(s_0)] = \mathbb{E}[Q^\star(s_0)]$, that is, it is optimal for a given initial state distribution. A policy $\pi$ is *uniformly optimal* if it satisfies $Q^\pi = Q^\star$, that is, it performs optimally for all starting states. An important theorem in the theory of MDPs, sometimes known as the fundamental theorem of MDPs, characterizes all uniformly optimal policies of an MDP.

**Theorem 8** (Fundamental Theorem of MDPs) *For every finite MDP, a policy $\pi : \mathcal{S} \to \mathcal{A}$ is uniformly optimal if and only if it is greedy w.r.t. $Q^\star$, that is, $\mathbb{E}_{\mathbf{a} \sim \pi(s)}[Q^\star(s, \mathbf{a})] = \max_a Q^\star(s, a)$ for all $s \in \mathcal{S}$.*

See Szepesvári (2023) for a proof.

# B Comparison of Constraints, Penalties, and Lexicographic Optimization

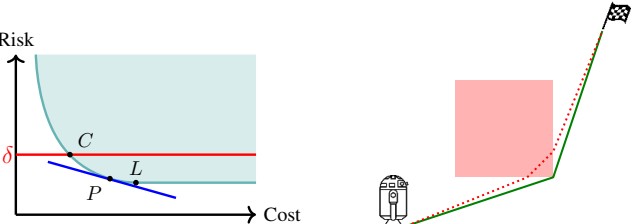

Figure 3: **(Left)** A set of policies $x$ (light cyan region), plotted by their $R, C$ values. The solutions of the constrained (C), penalty (P), and lexicographic (L) methods are indicated. The lowest-risk (safest) policy is given by (L). **(Right)** The agent aims to reach the target. The cost $C$ is the path length, and the risk $R$ is the portion of the path that lies within the unsafe (red) region.

In this section, we compare lexicographic optimization to the constrained approach and the penalty approach. We introduce a notation restricted to this section: $R$ denotes risk (the negation of the probability of safety), and $C$ denotes cost (the negation of the expected utility). The optimization problems corresponding to the constrained (C), penalty (P), and lexicographic (L) approaches are as

follows:

$$\min_{x} C(x), \qquad\qquad\qquad \text{subject to } R(x) \leq \delta \qquad\qquad (C)$$

$$\min_{x} C(x) + \lambda R(x) \qquad\qquad\qquad\qquad\qquad (P)$$

$$\mathrm{lex}\min_{x} \, (R(x), C(x)) \qquad\qquad\qquad\qquad\qquad (L)$$

The left part of Fig. 3 illustrates the optima of the different objectives over a two-dimensional set. The lexicographic minimizer yields the lowest-risk solution.

The right part of Fig. 3 illustrates a path optimization problem where $R$ and $C$ are defined as functions of the path $x$. The lexicographic minimizer is given by (L) and corresponds to the green path. The penalty method (P) leads to the dotted red path, which is shorter but has a nonzero $R(x)$. Either path can correspond to the constrained approach (C), depending on the value of $\delta$. Using hard constraints with a nonzero safety threshold or the penalty method results in a path that spends a nonzero amount of time in the unsafe region to reach the goal faster, resulting in the dotted red path. Using either hard constraints with a safety threshold of zero or lexicographic optimization results in the green path, which is optimal.

## C  Algorithmic Considerations for LMDPs

In MDPs, the optimal Q-value function $Q^{\star}(s, a)$ satisfies the *Bellman optimality equation*:

$$Q^{\star}(s, a) = \mathbb{E}\left[ R(s, a, \mathbf{s}') + \gamma \max_{a \in \mathcal{A}} Q^{\star}(s', a) \,\Big|\, s, a \right], \qquad (20)$$

where $R(s, a, s') \coloneqq \mathbb{E}[\mathbf{r} \mid s, a, s']$ is the expected reward from the transition $(s, a, s')$. The same is true for LMDPs, where $Q^{\star}(s, a)$ is a vector, $\gamma$ is a matrix, and the maximization in Eq. (20) is lexicographic.

The (uniformly) optimal policy $\pi^{\star}$ can then be derived from the optimal Q-value function as:

$$\pi^{\star}(s) = \operatorname*{argmax}_{a \in \mathcal{A}} Q^{\star}(s, a). \qquad (21)$$

When $\gamma < 1$, one can obtain $Q^{\star}$ by iterating the Bellman optimality equation until convergence. This algorithm is known as Q-value iteration.

---

**Algorithm 1** $\tau$-Approximate Q-Learning for LMDP

---

1: **Input:** LMDP with $d$-dimensional reward, slack $\tau \in \mathbb{R}_+^d$
2: **Initialize:** $Q(s, a) \leftarrow 0$ for all $s \in \mathcal{S}, a \in \mathcal{A}$; step size $\alpha \leftarrow 0.1$; $\epsilon \leftarrow 0.1$
3: Sample initial state $s \sim P_0$
4: **repeat**
5: $\quad \mathcal{A}_0^{\star}(s) \leftarrow \mathcal{A}$
6: $\quad$ **for** $i = 1$ **to** $d$ **do**
7: $\quad\quad \mathcal{A}_i^{\star}(s) \leftarrow \{a \in \mathcal{A}_{i-1}^{\star} \mid Q_i(s, a) \geq \max\limits_{a' \in \mathcal{A}_{i-1}^{\star}(s)} Q_i(s, a') - \tau_i\}$
8: $\quad$ **end for**
9: $\quad$ Sample action $a \sim \epsilon\, \mathrm{Uniform}(\mathcal{A}) + (1 - \epsilon)\, \mathrm{Uniform}(\mathcal{A}_d^{\star}(s))$
10: $\quad$ Execute $a$, observe next state $s'$ and reward vector $R \in \mathbb{R}^d$
11: $\quad$ **for** $i = 1$ **to** $d$ **do**
12: $\quad\quad Q_i(s, a) \leftarrow (1 - \alpha)Q_i(s, a) + \alpha\big(R_i + \gamma \max\limits_{a' \in \mathcal{A}_{i-1}^{\star}(s')} Q_i(s', a')\big)$
13: $\quad$ **end for**
14: $\quad s \leftarrow s'$
15: **until** $\tau$-convergence of $Q$
16: **Return:** $\pi(s) \leftarrow$ any $a \in \mathcal{A}_d^{\star}(s)$

---

Let $n$ be the number of state-action pairs. In an MDP, one can then think of the Q-value function as a point in $n$-dimensional space. The Q-value iteration algorithm converges because one can show

that the Bellman optimality equation is a contraction mapping in this space. The Banach fixed point theorem then implies convergence to a fixed-point.

However, the same argument fails for LMDPs because the Bellman optimality equation is not necessarily a contraction mapping. To obtain convergence one must accept some slack $\tau$ and take values that are closer than $\tau$ as being equal. This leads to Algorithm 1.

# D   Proofs

## D.1   Proof of Theorem 3

> **Theorem 3** (Sequential Lexicographic Expected Utility Theorem) *A relation $(\succsim, \Delta(\mathcal{E}^*))$ satisfies Completeness, Transitivity, Independence, and Memorylessness if and only if there exist $d \in \mathbb{N}$, a $d$-dimensional linear lexicographic utility function $u$ with $u(\varepsilon) = \mathbf{0}$, rewards $r : \mathcal{E} \to \mathbb{R}^d$, and reward multipliers $\Gamma : \mathcal{E} \to \mathcal{L}_+^{d \times d} \cup \{\mathbf{0}\}$ such that*
>
> $$u(e \cdot \tau) = r(e) + \Gamma(e)u(\tau), \tag{9}$$
>
> *for all $e \in \mathcal{E}$ and $\tau \in \mathcal{E}^*$.*

*Proof.* ($\Rightarrow$) By the lexicographic expected utility theorem (Theorem 2) there exist $d \in N$ and a $d$-dimensional linear lexicographic utility function $u$. Among the possible utility functions we pick one such that $u(\varepsilon) = \mathbf{0}$. Now, by Memorylessness, preferences are either retained when an event $e$ is prepended to lotteries, or all lotteries become indifferent. We handle the two cases below.

**1.**   If $e$ retains preferences, then the relation $(\succsim, \Delta(\mathcal{E}^*))$ is isomorphic to $(\succsim, e \cdot \Delta(\mathcal{E}^*))$, implying that their corresponding utility functions must be related by the uniqueness condition of Theorem 2. That is, for all $\tau \in \mathcal{E}^*$, $u(e \cdot \tau) = Au(\tau) + b$, where $A \in \mathcal{L}_+^{d \times d}$ and $b \in \mathbb{R}^d$. Let's name the corresponding $A$ and $b$ of $e$ as $\Gamma(e)$ and $r(e)$ respectively.

**2.**   If $e$ makes future lotteries indifferent, we must have for all $\tau \in \mathcal{E}^*$, $u(e \cdot \tau) = b$, where $b \in \mathbb{R}^d$. Again, we name the corresponding $b$ of $e$ as $r(e)$ and, to match the format of the previous case, we let $\Gamma(e)$ be the zero matrix $\mathbf{0}$.

In summary, for all $e \in \mathcal{E}$ and $\tau \in \mathcal{E}^*$, utilities are of the form

$$u(e \cdot \tau) = r(e) + \Gamma(e)u(\tau), \tag{22}$$

where $r : \mathcal{E} \to \mathbb{R}^d$ and $\Gamma : \mathcal{E} \to \mathcal{L}_+^{d \times d} \cup \{\mathbf{0}\}$.

($\Leftarrow$) By the lexicographic expected utility theorem (Theorem 2) the relation $(\succsim, \Delta(\mathcal{E}^*))$ corresponding to $u$ satisfies Completeness, Transitivity, and Independence. It remains to show that $\succsim$ satisfies Memorylessness.

Let $e \in \mathcal{E}$. We consider two cases:

**1.**   If $\Gamma(e) = \mathbf{0}$, then by Eq. (9), $u(e \cdot \tau) = r(e)$ for all $\tau \in \mathcal{E}^*$, so $u(e \cdot p)$ is constant across all lotteries $p \in \Delta(\mathcal{E}^*)$, implying $e \cdot p \approx e \cdot q$ for all lotteries $p, q$.

**2.**   If $\Gamma(e) \in \mathcal{L}_+^{d \times d}$, then for all lotteries $p, q \in \Delta(\mathcal{E}^*)$,

$$
\begin{aligned}
& e \cdot p \succsim e \cdot q \\
\Longleftrightarrow \ & u(e \cdot p) \geq_{\text{lex}} u(e \cdot q) && (u \text{ is a lexicographic utility function}) \\
\Longleftrightarrow \ & r(e) + \Gamma(e)u(p) \geq_{\text{lex}} r(e) + \Gamma(e)u(q) && (\text{Eq. (9)}) \\
\Longleftrightarrow \ & u(p) \geq_{\text{lex}} u(q) && (u \mapsto \Gamma(e)u + r(e) \text{ preserves } \geq_{\text{lex}}) \\
\Longleftrightarrow \ & p \succsim q. && (u \text{ is a lexicographic utility function})
\end{aligned}
$$

In both cases, the preference relation satisfies Memorylessness. $\qquad\square$

## D.2 Proof of Theorem 4

> **Theorem 4** (Discounted Lexicographic Expected Utility Theorem) *A relation $(\succsim, \Delta(\mathcal{E}^*))$ satisfies Completeness, Transitivity, Independence, and Temporal $\gamma$-Indifference if and only if there exist $d \in \mathbb{N}$, a d-dimensional linear lexicographic utility function $u$ with $u(\varepsilon) = \mathbf{0}$, and rewards $r : \mathcal{E} \to \mathbb{R}^d$ such that*
>
> $$u(e \cdot \tau) = r(e) + \gamma u(\tau), \tag{10}$$
>
> *for all $e \in \mathcal{E}$ and $\tau \in \mathcal{E}^*$.*

*Proof.* ($\Rightarrow$) By the lexicographic expected utility theorem (Theorem 2) there exist $d \in N$ and a $d$-dimensional linear lexicographic utility function $u$. Among the possible utility functions we pick one such that $u(\varepsilon) = \mathbf{0}$. Now, letting $\tau_1 = \tau$ and $\tau_2 = \varepsilon$ in Temporal $\gamma$-Indifference we get that for all $e \in \mathcal{E}$ and $\tau \in \mathcal{E}^*$,

$$\frac{1}{\gamma+1}(e \cdot \tau) + \frac{\gamma}{\gamma+1}(\varepsilon) \approx \frac{1}{\gamma+1}(e) + \frac{\gamma}{\gamma+1}(\tau)$$

$$\iff u\left(\frac{1}{\gamma+1}(e \cdot \tau) + \frac{\gamma}{\gamma+1}(\varepsilon)\right) = u\left(\frac{1}{\gamma+1}(e) + \frac{\gamma}{\gamma+1}(\tau)\right) \quad \text{($u$ is a utility function)}$$

$$\iff \frac{1}{\gamma+1}u(e \cdot \tau) + \frac{\gamma}{\gamma+1}u(\varepsilon) = \frac{1}{\gamma+1}u(e) + \frac{\gamma}{\gamma+1}u(\tau) \quad \text{($u$ is linear)}$$

$$\iff u(e \cdot \tau) = u(e) + \gamma u(\tau) \quad \text{($u(\varepsilon) = \mathbf{0}$).}$$

If we let $r(e) := u(e)$, then we have our result.

($\Leftarrow$) By the lexicographic expected utility theorem (Theorem 2) the relation $(\succsim, \Delta(\mathcal{E}^*))$ corresponding to $u$ satisfies Completeness, Transitivity, and Independence. It remains to show that $\succsim$ satisfies Temporal $\gamma$-Indifference.

For all $e \in \mathcal{E}, \tau_1 \in \mathcal{E}^*, \tau_2 \in \mathcal{E}^*$ we have

$$\frac{1}{\gamma+1}u(e) + \frac{\gamma}{\gamma+1}(u(\tau_1) + u(\tau_2)) = \frac{1}{\gamma+1}u(e) + \frac{\gamma}{\gamma+1}(u(\tau_1) + u(\tau_2))$$

$$\iff \frac{1}{\gamma+1}(u(e) + \gamma u(\tau_1)) + \frac{\gamma}{\gamma+1}u(\tau_1) = \frac{1}{\gamma+1}(u(e) + \gamma u(\tau_2)) + \frac{\gamma}{\gamma+1}u(\tau_1)$$

$$\iff \frac{1}{\gamma+1}u(e \cdot \tau_1) + \frac{\gamma}{\gamma+1}u(\tau_1) = \frac{1}{\gamma+1}u(e \cdot \tau_2) + \frac{\gamma}{\gamma+1}u(\tau_1) \quad \text{(Eq. (10))}$$

$$\iff u\left(\frac{1}{\gamma+1}(e \cdot \tau_1) + \frac{\gamma}{\gamma+1}(\tau_1)\right) = u\left(\frac{1}{\gamma+1}(e \cdot \tau_2) + \frac{\gamma}{\gamma+1}(\tau_1)\right) \quad \text{($u$ is linear)}$$

$$\iff \frac{1}{\gamma+1}(e \cdot \tau_1) + \frac{\gamma}{\gamma+1}(\tau_1) \approx \frac{1}{\gamma+1}(e \cdot \tau_2) + \frac{\gamma}{\gamma+1}(\tau_1) \quad \text{($u$ is a utility)}$$

$\square$

## D.3 Proof of Lemma 1

**Lemma 1.** *For all $p, q \in \Delta(\mathcal{O})$ and all $\alpha, \beta \in [0, 1]$,*

$$[\alpha, p] \succsim [\beta, q] \iff (\alpha > \beta) \text{ or } (\alpha = \beta \text{ and } p \succsim q). \tag{12}$$

*Proof.* We separate the space of possibilities into three cases:

1. If $\alpha > \beta$, then $[\alpha, p]$ has greater probability of avoiding $o^\dagger$. By Safety First, this implies $[\alpha, p] \succ [\beta, q]$, hence $[\alpha, p] \succsim [\beta, q] \iff$ True.

2. If $\alpha = \beta$, then by Independence, $[\alpha, p] \succsim [\alpha, q] \iff p \succsim q$.

3. If $\alpha < \beta$, then $[\alpha, p]$ has less probability of avoiding $o^\dagger$. By Safety First, this implies $[\alpha, p] \prec [\beta, q]$, hence $[\alpha, p] \succsim [\beta, q] \iff$ False.

Putting it all together, we get:

$$[\alpha, p] \succsim [\beta, q] \iff (\alpha > \beta \text{ and True}) \text{ or } (\alpha = \beta \text{ and } p \succsim q) \text{ or } (\alpha < \beta \text{ and False}).$$

This proves the desired equivalence. $\qquad\square$

### D.4 Proof of Theorem 5

**Theorem 5** (Lexicographic Expected Utility Theorem with a Single Unsafe Utility) *A relation* $(\succsim, \Delta(\mathcal{O}))$ *satisfies Assumption 1 if and only if there exist a linear utility function* $u' : \Delta(\mathcal{O}_{\mathrm{safe}}) \to \mathbb{R}$ *for* $(\succsim, \Delta(\mathcal{O}_{\mathrm{safe}}))$ *and a 2-dimensional linear lexicographic utility function* $u : \Delta(\mathcal{O}) \to \mathbb{R}^2$ *such that for all* $o \in \mathcal{O}$,

$$u(o) = \begin{cases} (0, u'(o)) & o \in \mathcal{O}_{\mathrm{safe}}, \\ (-1, 0) & o \approx o^\dagger. \end{cases} \tag{13}$$

*Moreover,* $u$ *is unique up to transformations of the form* $u \mapsto Au + b$, *where* $A \in \mathcal{L}_+^{2 \times 2}$ *and* $b \in \mathbb{R}^2$.

*Proof.* Given that $u'$ is linear, we uniquely extend Eq. (13) into a *linear* utility function on the entire domain. For all $p \in \Delta(\mathcal{O}_{\mathrm{safe}})$ and $\alpha \in [0, 1]$,

$$u([\alpha, p]) = (\alpha - 1, \alpha u'(p)). \tag{23}$$

($\Rightarrow$) The relation $(\succ, \Delta(\mathcal{O}_{\mathrm{safe}}))$ satisfies the vNM axioms so the vNM theorem implies the existence of a linear utility function $u'$. Showing that the function $u$ of Eq. (23) is a linear lexicographic utility function is a simple application of Lemma 1.

($\Leftarrow$) It is straightforward to check that preferences induced by the given $u$ satisfy axioms in the way described in the theorem.

Uniqueness: Let $o^+ \succ o^-$ be a best and worst outcome in $\Delta(\mathcal{O}_{\mathrm{safe}})$ respectively. Specifying the utility of $o^-$, $o^+$, and $o^\dagger$ uniquely specifies the entire utility function. There are 5 degrees of freedom because we must have $u_1(o^+) = u_1(o^-)$. These 5 degrees of freedom correspond to the 5 degrees of freedom in $A$ and $b$. Positivity of the diagonal of $A$ stems from the requirements that $u_2(o^+) > u_2(o^-)$ and $u_1(o^+) > u_1(o^\dagger)$. It is not hard to verify that $Au + b$ is a also a linear lexicographic utility function for $(\succsim, \Delta(\mathcal{O}))$. $\qquad\square$

### D.5 Proof of Theorem 6

**Theorem 6** (Sequential Lexicographic Expected Utility Theorem with a Single Unsafe Utility) *A relation* $(\succsim, \Delta(\mathcal{E}^*))$ *satisfies Assumption 1 and Memorylessness if and only if there exist rewards* $r : \mathcal{E} \to \mathbb{R}$, *reward multipliers* $\gamma : \mathcal{E} \to \mathbb{R}_+$, *and 2-dimensional linear lexicographic utility function* $u : \Delta(\mathcal{E}^*) \to \mathbb{R}^2$ *satisfying* $u(\varepsilon) = \mathbf{0}$ *and*

$$u(e \cdot \tau) = \begin{cases} (0, r(e)) & e \in \mathcal{E}_{\mathrm{term}} - \mathcal{E}^\dagger \\ (-1, 0) & e \in \mathcal{E}^\dagger \\ (u_1(\tau), (1 + u_1(\tau))r(e) + \gamma(e)u_2(\tau)) & \textit{otherwise} \end{cases} \tag{14}$$

*for all events* $e \in \mathcal{E}$ *and sequences of events* $\tau \in \mathcal{E}^*$.

*Proof.* By Theorem 5 there exists a 2-dimensional linear utility function $u : \mathcal{E}^* \to \mathbb{R}^2$ where the utility of any event that is indifferent to $o^\dagger$ is $(-1, 0)$ and the utility of every other event is in $\{0\} \times \mathbb{R}$. We let $\mathcal{U} := \{(-1, 0)\} \cup (\{0\} \times \mathbb{R})$ be this set of possible utilities.

By Theorem 3, there exist rewards $\tilde{r} : \mathcal{E} \to \mathbb{R}^2$ and reward multipliers $\tilde{\Gamma} : \mathcal{E} \to \mathcal{L}_+^{2 \times 2} \cup \{\mathbf{0}\}$ such that $u(\varepsilon) = \mathbf{0}$ and, for all $e \in \mathcal{E}, \tau \in \mathcal{E}^*$, utilities are of the form

$$u(e \cdot \tau) = \tilde{r}(e) + \tilde{\Gamma}(e)u(\tau). \tag{24}$$

Additionally, for all $x \in \mathcal{U}, e \in \mathcal{E}$, we must have

$$\tilde{r}(e) + \tilde{\Gamma}(e)x \in \mathcal{U}. \tag{25}$$

Eq. (25) puts a constraint on $\tilde{r}$ and $\tilde{\Gamma}$. Letting $x = (0,0)$ we conclude that $\tilde{r}(e) \in \mathcal{U}$. If $\tilde{\Gamma}(e) \neq \mathbf{0}$ ($e$ is non-terminal), then letting $x = (-1, 0)$ we get $\tilde{r}_1(e) - \tilde{\Gamma}_{1,1}(e) \in \{-1, 0\}$. Since $\tilde{\Gamma}_{1,1}(e) > 0$ we conclude that $\tilde{r}_1(e) - \tilde{\Gamma}_{1,1}(e) = -1$, $\tilde{r}_1(e) = 0$, and $\tilde{\Gamma}_{1,1}(e) = 1$. Consequently, we must have that $\tilde{r}_2(e) - \tilde{\Gamma}_{2,1}(e) = 0$, or equivalently, $\tilde{\Gamma}_{2,1}(e) = \tilde{r}_2(e)$ for non-terminal events.

Finally, letting $\gamma(e) := \tilde{\Gamma}_{2,2}(e)$ and $r(e) := \tilde{r}_1(e)$ we get the desired result. □

## D.6 Proof of Theorem 7

> **Theorem 7** (Fundamental Theorem of LMDPs) *For every finite LMDP satisfying Assumption 2, a policy $\pi : \mathcal{S} \to \mathcal{A}$ is uniformly optimal if and only if it is greedy w.r.t. $Q^\star$, that is, $\mathbb{E}_{a \sim \pi(s)}[Q^\star(s,a)] = \text{lex max}_a Q^\star(s,a)$ for all $s \in \mathcal{S}$.*

*Proof.* The proof relies on the fundamental theorem of MDPs (Theorem 8). Let $Q_1^\pi$ be the first dimension of the Q-value function under policy $\pi$ and let $Q_1^\star(s,a) := \max_\pi Q_1^\pi(s,a)$ for all $s, a$. This maximum exists since $Q_1^\pi$ is bounded (due to $\Gamma_{1,1}(e) < 1$) and the space of policies of a finite MDP is compact. The fundamental theorem of MDPs says that a policy $\pi^\star$ is uniformly optimal if and only if it is greedy w.r.t. $Q_1^\star$, that is, $\pi^\star(s) \in \Delta(\text{argmax}_a Q_1^\star(s,a))$. These policies are essentially restricted to choosing an action from a restricted set of actions at each state given by $\text{argmax}_a Q_1^\star(s,a)$. We can therefore imagine a smaller MDP with this restricted action set at each state. All policies of this MDP are optimal w.r.t. the first dimension of utility.

By Theorem 3, the second dimension of utility satisfies $u_2(e \cdot \tau) = r_2(e) + \Gamma_{2,1}(e)u_1(\tau) + \Gamma_{2,2}(e)u_2(\tau)$. Since all policies in this MDP are optimal w.r.t. the first dimension of utility, $\mathbb{E}_\pi[u_1(\tau) \mid s, a]$ is fixed and equal to $\mathbb{E}[V_1^\star(\mathbf{s}') \mid s, a]$ for all policies $\pi$ of the second MDP. This fixed value can be placed into the reward function without affecting $Q_2^\pi$. As a result, the second MDP can be viewed as a scalar MDP with the following reward: $r_2(e) + \Gamma_{2,1}(e)\mathbb{E}[V_1^\star(\mathbf{s}') \mid s, a]$.

Since the space of policies of this smaller MDP is a compact subset of the original space of policies and $\Gamma_{2,2}(e) < 1$, we can again invoke the fundamental theorem of MDPs for $Q_2^\pi$. We continue like this for all $d$ dimensions of utility. The Q-value of the final space of optimal policies is $(Q_1^\star, ..., Q_d^\star)$ which lexicographically dominates all $Q^\pi$ and is thus uniformly optimal. □

