# OpenReview forum: "Beyond Scalar Rewards: An Axiomatic Framework for Lexicographic MDPs"
_NeurIPS.cc/2025/Conference — NeurIPS 2025 spotlight_

### Official Review · Reviewer_ii8W · 2025-06-16

**Clarity:** 2
**Significance:** 3
**Originality:** 3
**Rating:** 5
**Confidence:** 3

**Summary:**

This paper takes an axiomatic approach to define Lexicographic MDPs (LMDPs) and proves key properties that LMDPs satisfy. The resulting framework enjoys desirable features such as the existence of uniformly optimal deterministic policies, something not guaranteed in CMDPs. One limitation, as noted in Section 8, is the absence of algorithms for solving the proposed LMDP framework.

Though I scored 3, I'm willing to raise the score if my concerns are addressed.

**Questions:**

* In line 124, what are the "other methods" being referred to?
* In line 36:

> Another common and practical example is when the first priority is to achieve some goal while the second priority is to minimize the time that it takes. Such objectives are also reminiscent of Asimov’s Three Laws of Robotics (Asimov, 1942), where the highest priority objective is to “not injure a human being or, through inaction, allow a human being to come to harm.”

Why is minimizing the time to achieve a goal aligned with Asimov's example? The connection is not clear to me.

======

* What does the $Q^\star$ in Theorem 7 refer to? I think the $Q$ function is not defined for LMDPs.
* Do you conjecture that the proposed LMDP framework satisfies a Bellman-type equation? The existence of a deterministic and uniformly optimal policy seems to suggest that such an equation might exist, which could be useful for constructing a dynamic programming algorithm.

**Ethical Concerns:**

["NO or VERY MINOR ethics concerns only"]

**Final Justification:**

The authors explain the relationship between LMDPs and CMDPs, which addresses my main concern about the significance of the results.

Since other reviewers also raised questions about the distinction from CMDPs, I suggest including a discussion on CMDPs in the camera-ready version.

**Limitations:**

The limitations are well described in Section 8.

**Paper Formatting Concerns:**

No formatting concern

**Quality:**

3

**Strengths And Weaknesses:**

**Strengths:**

The paper is generally well-written, and the theoretical results seem solid.
While no algorithm is provided, I believe that this axiomatic framework lays a strong foundation for future research on LMDPs.

**Weaknesses:**

The significance of the results is not well articulated.

What specific types of problems can this framework model compared to existing ones, like CMDPs?
If the LMDP framework can represent the same kinds of problems as CMDPs while guaranteeing a uniformly optimal deterministic policy, that would be a nice contribution.
However, I could not find any concrete examples illustrating such cases in the paper.
Although Figure 1 illustrates an example of an LMDP, it is not clearly explained how this example differs from what can be modeled by existing frameworks.

Explicitly identifying problems that the proposed framework can capture but existing works cannot would strengthen the paper’s contribution.

---

> ### Author Rebuttal · Authors · 2025-07-30
>
> We thank the reviewer for the constructive criticism and thoughtful questions. We reply to the questions below.
>
> Q: What specific types of problems can this framework model compared to existing ones, like CMDPs?
>
> A: There are CMDP problems that cannot be modeled as LMDPs. For example, any CMDP problem that leads to non-uniform optimal policy or stochastic optimal policy (and these do exist).
>
> In general, an LMDP cannot be modeled as a CMDP. However, if we fix the starting state distribution and, given K priority levels, we know the optimal value of all but the lowest priority level – i.e., we know $V^\star_1, …, V^\star_{K-1}$ – then it is possible to turn the LMDP problem into a CMDP, by constraining these $K-1$ values with the known optimal values and optimizing the lowest-priority value – that is,
> $$
> \max \mathbb{E}[V_K(s_0)] \text{ subject to } \mathbb{E}[V_i(s_0)] ≥ V^\star_i \text{ for all } i \in {1, …, K-1}.
> $$
> Basically, the first $K-1$ optimization problems are turned into constraints similar to turning $\max_x f(x)$ into $\max_x 0 \text{ subject to } f(x) ≥ f^\star$. We believe this to be a rather unnatural way of specifying the problem.
>
> When we have a single unsafe outcome, CMDP is constraining the probability of this outcome, assuming access to optimal achievable probability threshold (i.e., $V^\star_1$) rather than optimizing it. We contend that this assumption (of access to optimal values) is a limitation of CMDP framework that LMDPs naturally address.
>
> Q1: In line 124, what are the "other methods" being referred to?
>
> A: We claim that vNM reward systems have advantages compared to “other”s. Another way of specifying rewards is to assign a reward of 1 to optimal actions and 0 to all other actions. It is not practical but is still a valid reward system. More generally, there are many reward functions that can lead to the same optimal policy. A simple case: there are 3 actions $a, b, c$. We prefer $a \succ b \succ c$ but we use a reward function that satisfies $r(a) > r(c) > r(b)$ – notice that $b$ and $c$ are swapped. The optimal action is the same but the reward function has some undesirable properties. Suppose the initial policy selects action $b$ and then we optimize it (w.r.t. rewards) to $c$ but, due to “bad” reward function, the policy actually gets worse. On the other hand, with vNM rewards, increasing expected reward is guaranteed to result in a more preferred policy. Ad-hoc reward shaping methods can also be considered to be part of the "other" we mentioned.
>
> The advantage of vNM reward systems being insensitive to environment mechanics is also nicely demonstrated in the example in the response to first Q of reviewer AHWB where we want rewards that work regardless of the shape of the maze and hazard probabilities.
>
> Q2: line 36?
>
> A: They are not connected. The writing should be changed to "... the time that it takes. **Lexicographic** objectives are also reminiscent of Asimov’s Three Laws of Robotics..."
>
> Q3: What does the Q* in Theorem 7 refer to?
>
> A: The definition of Q* is the same as in scalar-reward MDP with the difference that maximization is w.r.t. lexicographic ordering of vectors. We will revise the text to clarify this.
>
> Q4: Do you conjecture that the proposed LMDP framework satisfies a Bellman-type equation?
>
> A: A Bellman equation does indeed exist and it is the same as the one for scalar-reward MDPs: $Q(s, a) = \mathbb{E}[R(s, a) + \gamma \max_{a’} Q(s’, a’)]$ with the distinction that Q-values are vectors and maximization is w.r.t. lexicographic ordering of vectors. However, in the lexicographic case, the Bellman operator is no longer a contraction mapping, so iterating it does not give us an algorithm with convergence guarantees.

---

> > ### Comment · Reviewer_ii8W · 2025-08-01
> >
> > Thank you for your response. It addressed my concerns, and I have updated my score accordingly.
> >
> > Since other reviewers also raised questions about the distinction from CMDPs, I suggest including a discussion on CMDPs in the camera-ready version.

---

> > > ### Comment · Area_Chair_JM1e · 2025-08-07
> > > **CMDPs vs LMDPs.**
> > >
> > > I agree that a discussion of CMDPs vs LMDPs would be a good addition to the paper.

---

> > > > ### Author Response · Authors · 2025-08-07
> > > >
> > > > We agree and we will certainly add an extensive discussion section to the camera-ready version.

---

### Official Review · Reviewer_AHWB · 2025-06-28

**Clarity:** 4
**Significance:** 2
**Originality:** 2
**Rating:** 4
**Confidence:** 4

**Summary:**

The authors provide an extension of Markov Decision Processes (MDPs) that supports lexicographic objectives. They take an axiomatic approach by describing criteria over preferences between histories (completeness, transitivity, independence, memorylessness, and temporal $\gamma$-indifference) and providing a representation result in the form of a linear lexicographic utility function (theorem 3 and 4). They additionally propose a "safety first" axiom that leads to a 2-dimensional lexicographic utility (theorem 5) which they extend to the sequential setting (theorem 6). Finally, they prove a "fundamental theorem of LMDPs" (theorem 7) that shows a relationship between optimal policies and the (lexicographically) optimal value function, analogous to that of MDPs, but, notably, unlike constrained MDPs.

**Questions:**

1. Do standard MDP algorithms work with LMDPs? What changes have to be made?
2. Have the authors tested methods for solving LMDPs? Are there any challenges not found in the standard MDP setting?
3. Can we state more formally what classes of problems are solved by LMDPs but not CMDPs?

**Ethical Concerns:**

["NO or VERY MINOR ethics concerns only"]

**Final Justification:**

The authors clarified why certain things were not included in the paper. I am still concerned that the absence of an algorithm for solving LMDPs will limit their usefulness to the field.

**Limitations:**

yes

**Quality:**

4

**Strengths And Weaknesses:**

Strengths
- This is a high quality paper: the technical results are clearly formulated and appear correct to me.
- The paper is clearly written: the authors do a good job of building up the results and connecting their framework/results to existing framework/results

Weaknesses
- Although the significance of the paper is gestured towards by comparing LDMPs with CMDPs and some comparision was done in the Appendix, I felt that the authors did not convincingly demonstrate how LMDPs can be used in place of scalar MDPs or CMDPs. Axiomatic definition of new classes of MDPs is a valuable exercise, however, there are practical reasons why the community uses MDPs as a standard model of sequential decision-making (e.g., we have lots of algorithms for solving MDPs). Unfortunately, the authors do not do very much to convince readers of the practical significance of not using LMDPs. In particular, I would have liked to see an LMDP of some reasonable difficultly actually be solved (e.g., a tabular and/or deep RL setting), either using a version of an MDP algorithm or a new algorithm. Even better would be to see how the CMDP formulation of the same problem leads to issues that LMDPs avoid. Currently, the authors only present a very simple lexicographic planning problem in Figure 1 and one in the appendix.
-  Relatedly, the "single unsafe utility" formulation is interesting and seems promising, but this is a single specific case of the more general formulation that seems like a good parameterization but isn't empirically demonstrated.
- The results presented are somewhat original. As far as I'm aware, there is no axiomatic derivation of lexicographic MDPs in the literature. However, the theoretical results seem to follow naturally from combining well-established results (e.g., expected utility, lexicographic expected utility, Bowling et al (2023)'s results on settling the reward hypothesis). In the end, the results feel like a straightforward exploration of a new way to specify sequential decision-making problems, but don't make the case for using this approach over any other in practice.

---

> ### Author Rebuttal · Authors · 2025-07-30
>
> Q: Significance of LMDPs and comparison to MDP and CMDP?
>
> A: Let us provide an example. Consider the classic problem of a robot in a maze trying to escape. To incentivize the robot to escape as fast as possible we either assign a reward of -1 for each step or we use a discount factor < 1 and assign a positive reward for escaping. Now consider a maze with *hazards* that can possibly destroy the robot. We are first and foremost interested in maximizing the probability of a safe escape out of the maze. Our second priority is to do so as fast as possible. Now suppose we have to design a reward function for this task without having seen the maze, e.g., we don’t know how likely the hazards are to destroy the robot or how large the maze is. The previous approaches break down in this setup. For a scalar reward function we can always design a maze such that the robot takes a more hazardous path by making safe paths longer and making hazard probabilities smaller. To quote Rich Sutton: “Approximate the solution, not the problem.” Using discount factors or negative rewards would be an approximation of the problem in this case. The true problem is lexicographic in nature. The reward for a hazardous event is (-1, 0), the reward for a safe step is (0, -1), and the reward for escaping the maze is (0, 0). Moreover, to be able to use CMDPs for this problem, we need to know what the optimal hazard probability is in order to set a constraint on it. Additionally, changing the starting state to a point in the maze that has a different optimal hazard probability necessitates respecifying the CMDP. Working on a better approximation of the solution to LMDPs is an interesting direction that we leave for future work.
>
> Q1: Do standard MDP algorithms work with LMDPs? What changes have to be made?
>
> Q2: Have the authors tested methods for solving LMDPs? Are there any challenges not found in the standard MDP setting?
>
> A: Value iteration is not guaranteed to converge since the Bellman operator is not a contraction mapping in the lexicographic setting. Intuitively, we first need to find the set of optimal actions w.r.t. Q_1, then construct an MDP restricted to these actions, and then optimize w.r.t. Q_2. Since value iteration for Q_1 isn’t guaranteed to converge in any finite number of steps we might never get to the second stage. What is sometimes done is to have a tolerance threshold τ such that Q-values that differ less than τ are considered equal. τ can go to 0 during training. However, this is a somewhat ad-hoc approach and can be difficult to tune. A method of encouraging Q-values for different actions to be equal via some kind of regularization could be an interesting direction for future work. Similar issues appear in policy-based algorithms as well. This new open problem that we also briefly discuss in future works is also the reason we cannot have an empirical evaluation.
>
> Q3: Can we state more formally what classes of problems are solved by LMDPs but not CMDPs?
>
> A: Please see answer to first Q of reviewer ii8W.

---

> > ### Comment · Reviewer_AHWB · 2025-08-06
> >
> > I thank the authors for their responses to my comments.

---

> > > ### Comment · Area_Chair_JM1e · 2025-08-07
> > > **Is the lack of algorithms for LMDPs a blocker?**
> > >
> > > Hi reviewer AHWB,
> > >
> > > Thanks for writing a great review!
> > >
> > > After reading the author response, do you think that the lack of exact algorithms for solving LMDPs is a major problem? Or do you think the paper is OK without?
> > >
> > > Thanks,
> > >
> > > area chair

---

> > > > ### Author Response · Authors · 2025-08-07
> > > >
> > > > (left out by mistake from original rebuttal) We would like to thank the reviewer for their thoughtful comments and feedback.

---

> > > > ### Comment · Reviewer_AHWB · 2025-08-08
> > > >
> > > > Yes, I do think it is a major problem. Without any kind of algorithm or analysis of an algorithm, its unclear how anybody can build on this work. In their response, the authors outline an approximate algorithm that seems reasonable (normal value iteration also does not give the "exact" values in general), if they could include that in the paper (with some additional analysis) I'd be satisfied.

---

> ### Author Response · Authors · 2025-08-09
>
> Certainly. We will add discussion and analysis of approximate Q-learning for LMDPs:
>
>
> *$\tau$-Approximate Q-learning for LMDP*
>
> *Input*: LMDP with $d$-dimensional reward; threshold $\tau \in \mathbb{R}^d_+$
>
> 1. $Q(s, a) \leftarrow 0$ for all $s \in \mathcal{S}, a \in \mathcal{A}$
> 2. $\alpha \leftarrow 0.1$ # learning step-size
> 3. $\epsilon \leftarrow 0.1$ # $\epsilon$-greedy policy
> 4. Sample initial state $s \sim P_0$
> 5. Repeat:
> 6. $\quad$ Let $\mathcal{A}^\star_0(s) = \mathcal{A}$
> 7. $\quad$ Let $\mathcal{A}^\star_i(s) = \set{ a \ |\  Q_i(s, a) \geq \max_{a \in
> \mathcal{A}_{i-1}^\star} Q_i(s, a) - \tau_i }$ for all $i \in [d]$
> 8. $\quad$ Sample action $a \sim \epsilon\ \text{uniform}(\mathcal{A}) + (1-\epsilon)\ \text{uniform}(\mathcal{A}^\star_d(s))$
> 9. $\quad$ Take action $a$ and transition into state $s'$ and obtain reward vector $R \in \mathbb{R}^d$
> 10. $\quad$ $Q_i(s, a) \leftarrow (1-\alpha)Q_i(s, a) + \alpha (R_i + \gamma \max_{a'
> \in \mathcal{A}_{i-1}^\star(s')} Q_i(s', a'))$
> 11. Return $\pi(s) = a \in \mathcal{A}^\star_d(s)$

---

### Official Review · Reviewer_gQjL · 2025-07-03

**Clarity:** 4
**Significance:** 2
**Originality:** 3
**Rating:** 5
**Confidence:** 4

**Summary:**

This paper contributes to the theory of lexicographic Markov decision processes, where the objective is to maximize a d-dimensional vector lexicographically. The paper starts by giving a background on:
- von-Neumann-Morgenstern (vNM) expected utility theorem: only linear utility functions satisfy the four vNM axioms
- if we let go of the “continuity” axiom, Hausner 1954 characterization theorem shows that we’ll have a $d$-dimensional linear *lexicographic* utility function for some d
The RL/MDP theory has largely focused on vNM utilities, and this paper lays a more generalized theory for lexicographic MDPs.

Building on Hausner 1954, the paper proposes a memorylessness axiom (extension of Shakerinava and Ravanbakhsh, 2022), and uses this axiom along with three of the vNM axioms to prove a characterization of linear lexicographic utility functions for MDPs or sequential decision making. Similar to the discounted reward case with the vNM utilities, the paper defines a discounted reward version and proves a characterization using another axiom it introduces.

Next, as an example, the paper investigates a setting where there are unsafe outcomes with infinite cost, which breaks the continuity assumption. The paper proves that this cannot be captured by vNM utilities and requires a 2-dimensional linear lexicographic utility function. The paper extends this example to the sequential setting as well.

Finally, the paper proves that for lexicographic MDPs, greedy following the adapted version of Q-value function is optimal and a stationary deterministic optimal policy exists.

**Questions:**

None at the moment.

**Ethical Concerns:**

["NO or VERY MINOR ethics concerns only"]

**Limitations:**

Yes

**Quality:**

3

**Strengths And Weaknesses:**

Strengths:

- The paper is written and presented very nicely. I appreciate the clarity. Though, none of the proofs appear in the main body. Though, the paper mentions that a reader knowledgeable of the proofs follow a structure similar to that of the scalar-reward models.
- The axiomatic analysis and the characterizations of lexicographic utility functions for MDPs is new --- though lexicographic objectives in MDPs have been investigated before.
- I haven’t checked the proofs, but the logical path of the theorems building on top of each other seems very reasonable and sound.
- The example with the unsafe outcome nicely illustrates that one needs lexicographic optimization.

Weaknesses:
I don’t see any critical problems, and I appreciate how clean everything is presented in the paper --- though, I haven’t checked any of the proofs yet.

I’m not currently giving a higher score mainly due to that, as I understand, the paper also fairly admits that many of the proofs and ideas are taken from prior work but nicely put together in this work for the sequential decision making setting.
The paper itself also mentions the algorithmic/computational challenges of solving lexicographic MDPs given the existing literature.
It would’ve been ideal to see some new (algorithmic/computational) insight taken away from the paper that can be used to devise new algorithms or perhaps solve or better formalize new problems. Though, I think there is value in the characterizations and the defined axioms.

---

> ### Author Rebuttal · Authors · 2025-07-30
>
> We thank the reviewer for their positive evaluation of our work. Please see the other responses for some arguments in favor of the significance of our work.

---

> ### Comment · Area_Chair_JM1e · 2025-08-07
> **Author response.**
>
> Hi reviewer gQjL,
>
> Have you reads the other reviews (and the author responses)?
> Has your opinion on the significance of the work changed?
>
> Thanks,
>
> area chair

---

> > ### Comment · Reviewer_gQjL · 2025-08-08
> >
> > Thank you for the response.
> >
> > I have read the other reviews and discussions. I agree with Reviewer 7CHw’s and Reviewer ii8W’s comments on the need for stronger comparisons to prior work and a clearer articulation of when LMDPs should be used over CMDPs. These additions would improve the paper. I also acknowledge that the lack of algorithmic development limits immediate applicability, and the paper would be much stronger with such results. Based on the points raised by other reviewers, my view of the significance has shifted slightly lower, but I still believe the theoretical characterization and axiomatic development are valuable in their own right and can serve as a foundation for future algorithmic work. My stance remains between borderline accept and accept, leaning toward accept.

---

### Official Review · Reviewer_7CHw · 2025-07-06

**Clarity:** 3
**Significance:** 2
**Originality:** 2
**Rating:** 4
**Confidence:** 3

**Summary:**

This paper studies reinforcement learning (i.e. sequential decision-making) under lexicographic preferences functions. The authors introduce the Lexicographic MDP (LMDP), and their contributions are to extend classical expected utility theory of von Neumann & Morgenstern and of Hausner to the sequential setting. Under one additional axiom that they argue is natural for the sequential setting (Memorylessness), they establish an analogue of Hausner's lexicographic expected utility theorem -- this gives rise to vector-valued utility functions taking a recursive form similar to the classical value functions. The authors then extend this to the setting of sequential decision-making with safety constraints, establishing a "safe" version of the lexicographic relations with an additional axiom (Safety First). It is shown that LMDPs with the Safety First axiom (and one additional technical axiom related to discount factors) have a unique globally optimal policy, similar to MDPs and in contradistinction with the Constraint MDPs.

**Questions:**

1. Differentiation from Prior Work: Can you provide a detailed technical comparison explaining how your contributions differ from these prior results beyond the tweaking of the axioms? What specific novel insights does your work provide that these earlier papers did not establish?
2. Scope of Safety Applications: Can you elaborate on the types of AI safety problems where your framework would be most appropriate and when one would choose this formalism over the Constrained MDP (or vice-versa)?

**Ethical Concerns:**

["NO or VERY MINOR ethics concerns only"]

**Limitations:**

Yes.

**Paper Formatting Concerns:**

None.

**Quality:**

3

**Strengths And Weaknesses:**

This is a theoretically interesting paper on the foundational aspects of decision-making with utilities. The motivation for studying this setting is the case when different vector-value outcomes can not be compared in a scalar fashion, i.e. there exists a strict ordering between the indices of the vector-valued outcome (maximizing the first index of the outcome takes highest precedence over any other index). The main application area that I can see is for providing a different formalism for safe decision-making. Accordingly, the authors study this case in the paper, with the axiom that there exists a single "infinitely bad" outcome that must be avoided at all costs.

The axioms the authors introduce are natural. The writing and presentation itself is very clear in the main text. One suggestion would be to review the abstract: it is vague does not really convey or accurately discuss the technical contributions of the paper.

As for what's missing:
- It would have been of high value to provide more discussion and perhaps some examples on how this "safe lexicographic decision-making" formalism differs from the Constrained MDP. Namely, is there a toy problem or perhaps a real-life example where one should use this formalism as opposed to the Constraint MDP? The authors do establish that they differ on at least one technical level (behaviours of their optimal policies), but this does not do enough to separate the two settings.
- Beyond this, as the authors acknowledge, it would have been interested to have some algorithmic developments (how to plan/solve the value/utility functions whose structure they have spent time working out).
- Lastly, there seems to be some missing discussion on how this work differs from related work *on a technical level*. The authors mention the works of (Shakerinava & Ravanbakhsh, 2022) and (Bowling et al., 2023) but not precisely how their results are similar or different. From a quick glance, the contents seem to be highly related. For example, Theorem 4.2 of (Shakerinava & Ravanbakhsh, 2022) studies the exact same axioms as Theorem 3 here except with the addition of the Continuity axiom, and derives an analogous conclusion. Similarly, Theorem 4.1 of (Bowling et al., 2023) is related to Theorem 4 here but without the sequential structure, and derives an analogous conclusion. This should be clearly discussed and at a glance reduces the novelty of the contributions in this paper.

I am leaving a weak accept for this paper, but would be happy to revise my score after the discussion if it turns out that I have been mistaken (or if the authors revise some of the discussion, if needed).

Smaller complaints:
- Please define lexmax_a in Theorem 7.

---

> ### Author Rebuttal · Authors · 2025-07-30
>
> We thank the reviewer for the detailed feedback and thoughtful questions. We will address the minor issues (abstract, define lex max). We respond to questions below.
>
> Q: Is there a [problem] where one should use this formalism as opposed to [CMDP]?
>
> A: We argue for an axiomatic approach to using objectives. If one accepts the independence axiom of von Neumann & Morgenstern as a natural axiom of rationality (which we the authors do), then one should use LMDPs (or MDPs if continuity is also accepted). For more comparison see also the response to first Q of reviewers ii8W and AHWB.
>
> Q: Comparison to (Shakerinava & Ravanbakhsh, 2022) and (Bowling et al. 2023)?
>
> A: This work is a natural next step for those papers. Is there a more general setup than vNM’s four axioms and what does that more general utility structure look like for MDPs? If one accepts the very natural Completeness, Transitivity, and Independence axioms then Hausner tells us that lexicographic rewards are the most general setting there is. We extend this result to MDPs and see that, similar to the scalar reward setting, lexicographic rewards take a simple recursive form.
>
> Q: Theorem 4.2 of (Shakerinava & Ravanbakhsh, 2022) vs. Theorem 3?
>
> A: The final recursive reward decomposition equation for both theorems look similar. The distinction is that in Theorem 3 rewards are vectors instead of scalars, and the factor Γ is a lower triangular matrix with positive diagonal instead of a positive scalar. A similar explanation goes for Theorem 4.1 of (Bowling et al., 2023) vs. Theorem 4. However, our Theorems 5, 6, 7 don’t have similar counterparts in these previous works.
>
> Q: What specific novel insights does your work provide that these earlier papers did not establish?
>
> A: Apart from Hausner’s work in 1954, before Bellman’s introduction of MDPs in 1957, recent prior works have largely ignored the lexicographic setting. It is only mentioned in passing in (Bowling et al. 2023). We provide a long overdue connection of Hausner’s work to MDPs and show that LMDPs (with a specific structure on the Γ matrix) provide the most general reward setting. Colloquially, the “correct” way to go beyond scalar-reward MDPs is to go to the LMDP setting (not, e.g., CMDPs).
>
> Q: AI Safety problem where this is useful?
>
> A: In AI safety, LMDPs can be appropriate when we introduce a highest-priority objective such as “do no harm.” In this setting, the agent must first minimize the probability of harm before optimizing any secondary performance criteria. This cannot be faithfully captured by scalar rewards. Also, the optimal harm threshold likely depends on the context so CMDPs cannot be used either. LMDPs offer a principled way to express such strict prioritization.

---

> > ### Comment · Reviewer_7CHw · 2025-08-07
> >
> > Dear authors,
> >
> > Thank you for your detailed reply.
> >
> > Regarding technical comparisons with prior work: I appreciate your clarification that some theorems represent natural extensions while others (Theorems 5, 6, 7) have no prior counterparts. However, my concern was about the insufficient discussion of these relationships in the paper itself. The current text doesn't adequately highlight how your work builds on and differs from recent related results. Could you add more explicit discussion comparing your contributions to Shakerinava & Ravanbakhsh (2022) and Bowling et al. (2023), and mentionning their influence?
> >
> > On the axiom-first approach: I recognize there may be philosophical differences in our perspectives. While I understand the theoretical elegance of deriving LMDPs from first principles, I remain more convinced by practical utility than axiomatic "naturalness." Your claim that LMDPs represent the "correct" way to extend beyond scalar rewards needs stronger justification beyond axiomatic arguments—particularly given that CMDPs have demonstrated practical success in many applications.
> >
> > On practical limitations: The AI safety example you provided (harm minimization with context-dependent thresholds) is helpful but limited. The paper would benefit from: (1) more concrete examples distinguishing when to use LMDPs vs. CMDPs, (2) discussion of computational approaches for solving LMDPs, and (3) acknowledgment of the practical challenges that the lack of algorithmic development presents.
> >
> > Given these ongoing concerns about practical utility and insufficient comparison with related work, I will maintain my borderline accept rating.

---

> > > ### Author Response · Authors · 2025-08-09
> > >
> > > > Could you add more explicit discussion comparing your contributions to Shakerinava & Ravanbakhsh (2022) and Bowling et al. (2023), and mentioning their influence?
> > >
> > > We will add a more explicit technical comparison to prior results (similar to our rebuttal response) in the camera-ready version.
> > >
> > > > While I understand the theoretical elegance of deriving LMDPs from first principles, I remain more convinced by practical utility than axiomatic "naturalness."
> > >
> > > Our work sits at a very fundamental level and discusses what the meaning of "practical utility" is theoretically. How to evaluate a system and how to optimize one are closely related. If one is not able to correctly evaluate a system then that makes it difficult to measure progress. In our framework, a system $A$ provides more practical utility compared to system $B$ if the distribution of outcomes that $A$ produces is preferred to the one produced by system $B$.
> > >
> > > If a user's notion of utility satisfies the independence axiom, then the CMDP objective most likely does not precisely capture the user's preferences and is not a precise way of evaluating that system. Furthermore, optimizing the CMDP objective is not guaranteed to result in a more preferred system.
> > >
> > > We tried to limit overlap with previous works but it can be useful to review how natural the independence axiom is. Suppose we have a biased coin that comes up heads with probability $p \in [0, 1)$ and suppose we want to choose between: (Choice 1) Toss the coin, if heads we get A, if tails we get B. (Choice 2) Toss the coin, if heads we get A, if tails we get C. Independence says that we prefer choice 1 iff we prefer B to C. That is, since the outcome for heads is the same in both cases, we only compare B and C to make our choice. Most would agree that this is a reasonable argument.
> > >
> > > We argue in this work that multi-criteria evaluation of systems must take a lexicographic form and we derive what such multi-criteria evaluations look like for MDPs. Evaluation and optimization go hand in hand and our work focuses on the theoretical foundations of evaluation in MDPs, and we leave algorithmic advances for optimization for future work.
> > >
> > > Even without a means of optimization, knowing that what we actually care about is a lexicographic metric is useful. If, in the process of whatever optimization algorithm we have access to (that optimizes some proxy), we manage to produce a few candidate policies, we can pick the best one among these by lexicographically comparing their utility vectors.
> > >
> > > > On practical limitations: The AI safety example you provided (harm minimization with context-dependent thresholds) is helpful but limited. The paper would benefit from: (1) more concrete examples distinguishing when to use LMDPs vs. CMDPs, (2) discussion of computational approaches for solving LMDPs, and (3) acknowledgment of the practical challenges that the lack of algorithmic development presents.
> > >
> > > (1) We will add a discussion on CMDP vs. LMDP in the camera ready version. (2) We will also describe and analyze a τ-approximate lexicographic Q-learning algorithm; because the lexicographic Bellman operator is non-contractive, we do not claim asymptotic exactness. (3) We acknowledge that designing convergent, scalable algorithms for LMDPs remains open; our contribution is to pin down the correct objective and its structural properties.
> > >
> > > We'd also like to explicitly state that we believe all the questions raised by the reviewers to be good and helpful and we will use the extra page in the camera-ready version to incorporate our responses into the paper.

---

> ### Comment · Area_Chair_JM1e · 2025-08-07
> **Are you happy with the rebuttal?**
>
> Hi reviewer 7CHw,
>
> Have you read the rebuttal? What are your thoughts?
>
> Thanks,
>
> area chair

---

### Decision · Program_Chairs · 2025-09-17

**Decision:**

Accept (spotlight)

**Comment:**

This paper provides an axiomatic derivation of lexicographic MDPs. Lexicographic MDPs are useful to represent certain types of preferences that classic MDPs are not expressive enough to capture.

The main strength is that lexicographic MDPs, while being in their early stages of development, are a promising modelling tool, particularly for problems involving safety.

There were a couple of minor weaknesses: novelty of proof techniques, lack of sufficient comparison with constrained MDPs and basic algorithmic tractability (i.e. lack of algorithms for solving lexicographic MDPs). These were largely addressed in the discussion.

The discussion focussed on the weaknesses of the paper as above.